Assessing arthropod biodiversity with DNA barcoding in Jinnah Garden, Lahore, Pakistan

http://orcid.org/0009-0004-0888-2148 Samreen Khush Bakhat 1 cool.khushi19@gmail.com
Manzoor Farkhanda 2
1 Department of Zoology, Lahore College for Women University, Lahore , Lahore , Pakistan
2 Department of Zoology, Minhaj University , Lahore, Punjab , Pakistan
Kandhro Abdul Hafeez
Electronic publication date: 2024 May 31
Publication date: 2024
Volume: 12
Electronic Location ID: e17420
Received 2023 Dec 29; Accepted 2024 Apr 28
Copyright: © 2024 Samreen and Manzoor
Copyright year: 2024
Copyright holder: Samreen and Manzoor
License: This is an open access article distributed under the terms of the Creative Commons Attribution License, which permits unrestricted use, distribution, reproduction and adaptation in any medium and for any purpose provided that it is properly attributed. For attribution, the original author(s), title, publication source (PeerJ) and either DOI or URL of the article must be cited.
License URL: https://creativecommons.org/licenses/by/4.0/

Keywords: Arthropods taxonomy, Biodiversity assessment, Barcode of Life Data System (BOLD), Barcode Index Numbers (BINs)

Funding: The authors received no funding for this work.

==============================
Previous difficulties in arthropod taxonomy (such as limitations in conventional morphological approaches, the possibility of cryptic species and a shortage of knowledgeable taxonomists) has been overcome by the powerful tool of DNA barcoding. This study presents a thorough analysis of DNA barcoding in regards to Pakistani arthropods, which were collected from Lahore’s Jinnah Garden. The 88 % (9,451) of the 10,792 specimens that were examined were able to generate DNA barcodes and 83% (8,974) of specimens were assigned 1,361 barcode index numbers (BINs). However, the success rate differed significantly between the orders of arthropods, from 77% for Thysanoptera to an astounding 93% for Diptera. Through morphological exams, DNA barcoding, and cross-referencing with the Barcode of Life Data system (BOLD), the Barcode Index Numbers (BINs) were assigned with a high degree of accuracy, both at the order (100%) and family (98%) levels. Though, identifications at the genus (37%) and species (15%) levels showed room for improvement. This underscores the ongoing need for enhancing and expanding the DNA barcode reference library. This study identified 324 genera and 191 species, underscoring the advantages of DNA barcoding over traditional morphological identification methods. Among the 17 arthropod orders identified, Coleoptera, Diptera, Hemiptera, Hymenoptera, and Lepidoptera from the class Insecta dominated, collectively constituting 94% of BINs. Expected malaise trap Arthropod fauna in Jinnah Garden could contain approximately 2,785 BINs according to Preston log-normal species distribution, yet the Chao-1 Index predicts 2,389.74 BINs. The Simpson Index of Diversity (1-D) is 0.989, signaling high species diversity, while the Shannon Index is 5.77, indicating significant species richness and evenness. These results demonstrated that in Pakistani arthropods, DNA barcoding and BOLD are an invaluable tool for improving taxonomic understanding and biodiversity assessment, opening the door for further eDNA and metabarcoding research.

Introduction

The problems with arthropod taxonomy are not localized; they exist worldwide. When employed extensively, traditional morphological techniques, which have been utilized for many years to identify species, present a number of challenges. First off, it can be challenging to develop consistent and trustworthy identification criteria due to the significant morphological variation that some species can show within their populations (Hebert et al., 2003). Furthermore, it can be particularly difficult to detect cryptic species—that is, animals that share physical features but have unique genetic characteristics—using conventional methods (Williams, Ormerod & Bruford, 2006). Moreover, phenotypic plasticity, the ability of physical traits to alter in response to environmental factors, is exhibited by many species. This might result in inconsistent morphological identification, since members of the same species may display distinct characteristics depending on their physical characteristics can change in response to environmental conditions. This can lead to inconsistencies in morphological identification, as individuals within the same species may exhibit different traits based on their environmental context (Moczek, 2010). The lack of qualified taxonomists exacerbates these problems by potentially impeding prompt and precise species identification. Examining preserved specimens is necessary for morphological identification; however, poor preservation practices, specimen damage, or insufficient preservation can mask or change key morphological characteristics, making identification more challenging (Cognato et al., 2020). Finally, it may become more difficult to differentiate between morphologically similar species based alone on morphology due to convergent evolution, which is fueled by comparable environmental forces and can result in the formation of similar features in various species (Montealegre et al., 2012).

This highlights the requirement for various methods to raise the effectiveness and precision of species identification. To get around these issues and increase precision and dependability, researchers commonly combine morphological identification with molecular methods like DNA barcoding (Seberg et al., 2003). DNA barcoding is a widely used method in many different fields, such as: phylogenetic studies (Hajibabaei et al., 2007), taxonomic analysis (Dewalt, 2011), looking at biodiversity of insect communities (Hlebec et al., 2022), examining genetic patterns (Zhou et al., 2010), phylogenetic analysis (Huang, Yang & Ke, 2016), and food authentication and safety (Dawan & Ahn, 2022). Through the examination of particular genetic markers, such as the cytochrome oxidase I (COI) gene, which is highly conserved among species and less susceptible to homoplasy, DNA barcoding has become an effective tool in species identification (Gonzalez et al., 2024; Antil et al., 2023). Moreover, this 658-base pair sequence, often referred to as the “DNA barcode,” acts as a unique marker for species identification due to its significant sequence variation, which helps differentiate species that are closely related (Jinbo, Kato & Ito, 2011).

DNA barcoding’s rapid adoption in modern biodiversity research (Hebert et al., 2003) has been powered by its use in specimen identification. This impressive efficacy of DNA barcoding in enabling thorough assessments of biodiversity is also demonstrated (Wilson et al., 2017; D’Souza et al., 2021; Shashank et al., 2022). The study of Wilkinson et al. (2017) shows that development of next-generation sequencing technologies has expedited the identification and discovery of previously unknown species, significantly increasing the speed and efficiency of DNA barcoding. Moreover, DNA barcoding has significantly advanced our comprehension of biological diversity by focusing on specific, consistent DNA sequences like the internal transcribed spacer (ITS) region. This method delivers high accuracy and dependability, even among species that are closely related, as evidenced by the research of Tyagi et al. (2019).

DNA barcodes are unique sequences that act like biological identification tags for species. These barcodes are central to the Barcode of Life Data System (BOLD), which is an open-access platform that simplifies the tasks of classifying species, identifying unknown specimens, and discovering new species (Hebert et al., 2003; Miller et al., 2016). BOLD is an abundant resource that provides a multitude of DNA barcode records from many taxonomic categories. By making a large database of barcode sequences easily accessible, this technology expedites the process of identifying species and enables scientists and researchers to compare and evaluate genetic data from different organisms (Ratnasingham & Hebert, 2007). BOLD is an essential informatics platform for biodiversity and evolutionary research, offering a user-friendly interface that simplifies the management and analysis of genetic data (Meiklejohn, Damaso & Robertson, 2019). BOLD integrates molecular, morphological, and distributional data, bridging gaps in bioinformatics and supporting global research collaborations. By adhering to stringent data standards, BOLD ensures the quality and reliability of genetic information, making it an invaluable resource for the scientific community (Ratnasingham & Hebert, 2007; Meiklejohn, Damaso & Robertson, 2019).

A Barcode Index Number (BIN) is a unique identification number assigned to each species. Its species’ unique DNA barcode sequence serves as the basis for its BIN. The cytochrome c oxidase subunit I (COI) gene, a standardized region of the genome with notable species variation, serves as the foundation for numerous studies. It is the starting point for delineating species as even a few hundred base pairs from this region are adequate to create distinct Barcode Index Numbers (BINs) (Ratnasingham & Hebert, 2013). As to the findings of Ratnasingham & Hebert (2013) each BIN is assigned to a particular species or genus, making the complex subject of taxonomy a little easier to navigate and comprehend.

BINs have vast scope beyond their significance in the discovery of new species. They are useful tools for tracking the locations of current species (Ren et al., 2018) and for estimating species abundances within large samples (Andújar, García-de la Torre & Rodríguez, 2018; Braukmann et al., 2019). It is similar to using a microscope to enlarge on the populations of various species concealed in large samples, allowing us to learn more about the complex web of life. In addition, the use of BINs for DNA barcoding has made it easier for researchers to examine museum collections and learn more about past biological assemblages (Pentinsaari et al., 2020). Furthermore, scientists can assess the degree of similarity or dissimilarity across those populations by comparing the BIN profiles of fauna from other locations and the world at large (Ashfaq et al., 2017), which advances our knowledge of trends in global biodiversity.

Keeping in view the importance of DNA barcoding, the current study significantly expands the scope of DNA barcoding for Pakistani arthropods, thereby advancing our understanding of the country’s taxonomic biodiversity and laying the foundation for future eDNA and metabarcoding investigations.

Materials and Methods

Sample collection and preparation

Sampling

Jinnah Garden is a public park covering approximately 16 hectares (around 39 acres) at coordinates 31.5533°N, 74.3306°E. Topographically, the park showcases a mix of flat and hilly landscapes, adorned with numerous walking paths and walkways that intersect throughout its expanse. The arthropod specimens were collected by using Malaise traps across 28 sites located on the flatlands of the park shown in Fig. 1 (Pentinsaari et al., 2020).

Figure 1 Jinnah Garden map showing 28 collection sites for arthropod specimens examined in this study.

The color of each site point indicate the number of specimens sampled. Map was generated in GIS-map satellite imagery.

The samplings were conducted over a period of thirty one (31) months (March 2019–September 2021). The specimens were collected on 10th, 20th and 30th day, with 10 days interval each and three trapping days per month. As sampling were made three times per month, here were a total of 93 sampling days (Kaczmarek, Entling & Hoffmann, 2022). There was a good turnover of arthropod species (41–50 specimens every 10th day) on seven sites with malaise traps in warmer and drier weather (Fig. 1) (Kirse et al., 2021). The collected specimens were euthanized in cyanide jars, placed in paper envelopes, and then relaxed, pinned, labeled, and stored at the Entomological Laboratory within Lahore College for Women University (Kaczmarek, Entling & Hoffmann, 2022).

Specimen identification

Specimen identifications were carried out till order level with valuable insights from literature (Askew & Barnard, 1988; Gibb & Oseto, 2006; Thyssen, 2010). To enhance accuracy, morphological identifications were cross-referenced whenever feasible by comparing the Pakistani specimens’ barcode records with pre-existing records on BOLD. The complete collected data along with images and specimen details were submitted to BOLD and can be accessed through the dataset “DS-GMPJA” Malaise trap Jinnah Garden Lahore on BOLD database (https://www.boldsystems.org/index.php/Public_BINSearch).

DNA barcoding/molecular analysis

A total of 10,792 insects were subjected to barcoding in Jinnah Garden, Lahore, following the established protocols (deWaard et al., 2019a, 2019b). Briefly, for larger specimens, a leg was carefully removed using sterile forceps and transferred to a well containing 30 ml of 95% ethanol. Smaller specimens, already on plates, were prepared for analysis, with vouchers retrieved after DNA extraction (Porco et al., 2010). At the Canadian Centre for DNA Barcoding (CCDB), we followed well-established procedures for DNA extraction, PCR amplification, and sequencing. These methods were described in previous publications (Ivanova, deWaard & Hebert, 2006; Hebert et al., 2018, deWaard et al., 2019b). Depending on the specific experiment, we used either 6 or 12 milliliters of material for the PCR reactions (Hebert et al., 2013).

Using an Applied Biosystems 3730XL DNA Analyzer and the BigDye Terminator Cycle Sequencing Kit (v3.1), specimens were subjected to Sanger sequencing. Afterward, CodonCode Aligner was used to assemble, align, and modify the sequences before being submitted to BOLD (Richterich, 2004). All DNA extracts are stored within the DNA archive facility at Centre for Biodiversity Genomics (CBG), Guelph, Canada.

Data analysis

The specimens of final dataset received BINs and taxonomy assignments according to the workflow proposed by deWaard et al. (2019b). This involved a two-step process, where first, the barcode data was uploaded onto BOLD, and then each record underwent taxonomic assignment and verification (deWaard et al., 2019b). Morphological study by taxonomic specialists was also conducted alongside the molecular analysis to enhance species delimitation. Prior studies have shown the benefit of integrating both approaches on Lepidoptera (Silva-Brandão, Lyra & Freitas, 2009), on Aranea (Blagoev et al., 2013) and on Hemiptera specimens collected from various sites in Pakistan (Naseem et al., 2019). These studies demonstrate the advantages of combining molecular and morphological techniques for accurate species identification. By following the approach outlined in the literature, only sequences that met the criteria of quality were either assigned to already existing BINs or used to create new ones (Ratnasingham & Hebert, 2013).

To delineate new Barcode Index Numbers (BINs), the protocol required adherence to stringent quality criteria. Eligibility for BIN classification required sequences to span at least 500 base pairs of the barcode region, specifically between positions 70 and 700 on the alignment of BOLD contain less than 1% ambiguous bases, and be devoid of stop codons or contamination indicators (Ratnasingham & Hebert, 2013). Additionally, sequences of shorter length (300–495 base pairs) that met the quality standards—lacking ambiguous bases and stop codons—and demonstrated high similarity to an established BIN were consolidated under the corresponding BIN (deWaard et al., 2019a). These sequences underwent analysis using the stand-alone version of the RESL algorithm (via the “Cluster Sequences” function on BOLD) and further authenticated by generating the “taxon ID tree” analysis on BOLD. Comprehensive BIN data, inclusive of specimen records and their images where available, are accessible through the BOLD interface at “DS-GMPJA” Malaise trap Jinnah Garden Lahore.

A BIN discordance analysis was employed to ascertain the proper BIN assignments within BOLD. Unassigned specimens underwent scrutiny via the BOLD Identification Engine (http://www.boldsystems.org/index.php/IDS_OpenIdEngine). Ensuing assignments underwent corroboration through the taxon ID tree to ensure accuracy. Sequences identified as contamination were consequently flagged, cataloged as such on BOLD, and excised from both the analysis and their associated BIN entries.

The BIN discordance report leveraged the comprehensive suite of functions within MS Excel to compute summary statistics. Furthermore, indices of species richness and evenness (such as Preston plot, Chao-1 index, Shannon Index, Simpson Index of Diversity) were evaluated for 8,451 specimens (that were assigned BINs) using the “Diversity measure” function provided by BOLD.

Results

DNA barcoding analysis of 10,792 specimens yielded successful results for 9,451 (88%), the remaining 12% were excluded from additional investigation because they were either unable to amplify or produced undesirable sequences (such as contamination, NUMTs, stop codons, or endosymbionts) (Files S1 and S2). Sequence recovery varied widely among orders with 100 or more specimens, ranging from 77% for Thysanoptera to 93% for Diptera. There was significant difference in the sequence recovery for the four orders of insects, which are the Coleoptera (83%), Hymenoptera (81%), Hemiptera (80%), and Lepidoptera (72%) (Table S1).

Among the 9,541 successfully barcoded records, 8,974 (95%) were assigned to BINs, leaving 477 records that did not qualify for BINs. These 477 sequences, not meeting BIN criteria, underwent analysis using the stand-alone version of the RESL algorithm (via the “Cluster Sequences” function on BOLD), revealing 386 Operational Taxonomic Units (OTUs), potentially representing distinct species. Of these, only 18 OTUs (encompassing 24 records) were free from contamination or stop codons (File S3). Each of these 18 OTUs was cross-referenced with the BOLD ID engine, revealing no matches to known BINs and suggesting they are novel to BOLD, as further supported by “taxon ID tree” analysis on BOLD (File S4).

The 8,974 barcodes successfully assigned were distributed across 1,361 BINs. Notably, 191 unique BINs (14%) were exclusively identified at the Jinnah Garden site, while the remaining 1,170 BINs (86%) were shared with other locations, both within and outside Pakistan. In terms of taxonomic classification, 98% of the barcodes (9,255) belonged to the Class Insecta, followed by Class Arachnida (99 barcodes, 1%), Class Collembola (91 barcodes, 0.96%), and Class Malacostraca (five barcodes, 0.05%). The Class Arachnida specimens were further categorized into four orders (Araneae, Mesostigmata, Sarcoptiformes, and Trombidiformes), encompassing 17 families, 14 genera, and nine species. Collembola included two orders (Entomobryomorpha and Symphypleona), with the former yielding three families, three genera, and two species. Malacostraca featured only the order Isopoda, with four barcodes across one species.

In the class Insecta, specimens were assigned across 10 orders, with 98% falling into 149 families (as detailed in Table 1). The majority (92%) belonged to three orders: Diptera (66%), Hymenoptera (16%), and Hemiptera (10%), as shown in Fig. 2. Other orders like Coleoptera, Lepidoptera, and Thysanoptera each had over 50 specimens, while Neuroptera, Odonata, Orthoptera, and Psocodea had fewer.

Table 1 Count of specimens having DNA barcode records from Jinnah Garden in Lahore, Pakistan, that belong to 17 orders.

Order	Specimens with barcodes	Specimens assigned to BINs (%)	BINs recovered	OTUs without BIN	Singleton BINs (%)	BINs assigned to family (%)	Families recovered	BINs assigned to genus (%)	Genera recovered	BINs assigned to species (%)	Species recovered	
Class Arachnida	
Araneae	36	59.89	15	1	64.29	85.71	11	71.43	10	28.57	4	
Mesostigmata	24	100.00	11	0	36.36	81.82	4	54.55	3	27.27	3	
Sarcoptiformes	3	100.00	1	0	0.00	0.00	0	0.00	0	0.00	0	
Trombidiformes	15	100.00	5	0	60.00	80.00	2	40.00	1	40.00	2	
Class Collembola	
Entomobryomorpha	89	97.75	12	0	33.33	83.33	3	41.67	3	16.67	2	
Symphypleona	2	50.00	1	0	100.00	0.00	0	0.00	0	0.00	0	
Class Malacostraca	
Isopoda	5	80.00	1	0		100.00	1	100.00	1	100.00	1	
Class Insecta	
Coleoptera	277	88.45	86	2	54.65	100.00	24	30.23	21	18.60	16	
Diptera	6,142	97.75	530	6	40.75	99.81	38	32.33	103	12.22	64	
Hemiptera	996	89.06	122	0	42.62	99.18	23	59.02	51	31.15	33	
Hymenoptera	1,547	91.79	490	8	57.76	97.55	39	31.22	89	7.55	37	
Lepidoptera	155	83.23	48	0	52.08	87.50	14	58.33	25	31.25	15	
Neuroptera	1	100.00	1	0	100.00	100.00	1	100.00	1	0.00	0	
Odonata	5	100.00	2	0	50.00	100.00	1	50.00	1	0.00	1	
Orthoptera	6	83.33	3	1	66.67	100.00	3	66.67	2	66.67	2	
Psocodea	34	88.24	9	0	44.44	77.78	4	33.33	2	22.22	2	
Thysanoptera	92	89.01	24	0	41.67	100.00	2	70.83	11	41.67	10	
Total	9,450	95%	1,361	18	49%	98%	170	37%	324	15%	191	
Note:

For every order, the number of families, genera, species, and BINs is given (Table S1).

Figure 2 Taxonomic assignment for 17 orders of four classes of Phylum Arthropoda.

Among 10,792 specimens, 97% of specimens (N = 10,448) were accompanied by images. Most sequences (95%) received a BIN assignment, cumulating in 1,361 BINs. Over half (51%) of the 1,361 BINs were represented by a minimum of two or more sequences, while the remaining 49% were represented by only a single specimen. Notably, the proportion of these single-specimen BINs exceeded 40% in the orders of Coleoptera, Diptera, Hemiptera, and Hymenoptera, with the highest occurrence in Hymenoptera (58%, N = 283). The assignment of BINs varied across different orders, with the order Araneae (Class Arachnida) showing an 82% success rate, and the order Entomobryomorpha (Class Collembola) achieving 85%. Within Class Insecta, the distribution of BIN assignments was as follows: Diptera and Hymenoptera both at 79%, Coleoptera at 75%, Lepidoptera at 64%, and Hemiptera at 52% (refer to Table 1). Together these five orders contributed to 94% of the BINs and 81% of the families identified (as shown in Table 1, Figs. 2 and 3).

Figure 3 BINs assignment for 17 orders of four classes of Phylum Arthropoda.

Regarding taxonomic resolution, 37% of BINs were identified to the genus level, and 15% to the species level. This led to the identification of 324 genera and 191 species (Table 1). In Class Insecta, a higher proportion of BINs were identified in order Diptera to the (Genus: 32%, N = 172; Species: 12%, N = 65) and Hymenoptera (Genus: 31%, N = 153; Species: 7%, N = 37). The representation of the 170 families was highly variable. Twenty-one families were represented by more than 100 specimens each, while 33 families had only a single specimen (Table 1). This variation was also observed in the number of BINs: 16 of the 21 families with more than 100 specimens had over 20 BINs, whereas 59 families had only one BIN. The families Ceratopogonidae (N = 1,489) and Chloropidae (N = 680) had the highest specimen counts in the order Diptera, Class Insecta. In terms of diversity, Cecidomyiidae (92 BINs) in order Diptera and Scelionidae (70 BINs) in order Hemiptera were the most diverse.

Figure 4 illustrates the diversity of BINs and the BIN-to-specimen ratio for the 21 families with more than 20 BINs. The ratio was highest for Scelionidae in Hemiptera (0.42) and lowest for Anthomyiidae and Ceratopogonidae in Diptera (0.03). Based on the Preston log-normal species distribution, species richness extrapolation indicates that a thorough examination of Jinnah Garden’s Malaise-trappable arthropod fauna might yield roughly 2,785 BINs, or almost twice as many as have been found. However, the Chao-1 Index estimates the species richness at 2,389.74 BINs. The Simpson Index of Diversity (1-D) is 0.989, indicating very high species diversity, and the Shannon Index is 5.77, suggesting high species richness and significant species evenness (Table 2, Fig 5) (File S5).

Figure 4 BIN: specimen ratio and BIN diversity for 21 insect families (of three orders) with more than 20 BINs each.

Table 2 Species richness estimates based on the abundance of 1,361 BINs encountered at various sites in Jinnah Garden, Lahore, Pakistan (File S5).

Specimens	BINs	Preston	Chao-1 index	Simpson index of diversity (1-D)	Shannon index	
8,974	1,361	2,785	2,389	0.98	5.77	

Figure 5 Preston plot with veil line (and species richness extrapolation) based upon the abundance data of 8,451 arthropods taxa which generated a sequence.

Discussion

Pakistan’s arthropod biodiversity has been estimated to contain anywhere from approximately 5,000 to 20,000 species, according to various sources (Hasnain, 1998; Ministry of Climate Change, Pakistan, 2019). Nevertheless, these previous estimates have been deemed insufficient by recent studies (Baig & Al-Subaiee, 2009; Rana et al., 2019). This research aimed to provide a more accurate assessment of regional fauna by utilizing DNA barcoding and the BIN system. A thorough examination was conducted on more than 10,792 specimens gathered from Jinnah Garden in Lahore, Pakistan. This exhaustive endeavor aimed to establish a comprehensive DNA barcode library of the region’s arthropod fauna.

Although the success rate (88%) for recovering DNA barcodes was good, it differed significantly between orders from 77% for Thysanoptera to 93% for Diptera (with more than 100 specimens). In other research, similar variance in barcode recovery across several arthropod taxa has been documented. The sequence recovery rates ranging from 75% to 80% has been reported in various insect orders (Park et al., 2011; Ashfaq et al., 2022). Moreover, similar trends of variation in sequence recovery rates in a broader scope has been reported across Canada, DNA barcoding was performed on 1,500,003 animal specimens across diverse taxonomic levels. Of these, 1,457,334 (97.2%) were assigned to a total of 64,264 BINs (deWaard et al., 2019a).

The 12% of specimens (1,341) were not assigned any barcode and excluded from further investigation. Factors contributing to this include failures in primer binding due to genetic variation or mismatches between primers and DNA (Elbrecht, Hebert & Steinke, 2018; Wilson et al., 2017). Co-amplification of pseudogenes may introduce errors (Leite, 2012), while endosymbionts like Wolbachia can hinder DNA extraction (Jones, Ghoorah & Blaxter, 2011; Smith et al., 2012). Recent speciation events and genetic similarity among closely related species pose challenges to species identification using a single barcode marker (Soria-Carrasco et al., 2014; Yasuda et al., 2015).

The combination of morphological examination and barcode matching on BOLDi (deWaard et al., 2019a, 2019b) proved highly effective in assigning BINs to an order level with 100% efficacy and 98% to the family level. However, just 37% of BINs were identified to the genus level, and 15% to the species level. This resulted in the identification of 324 genera and 191 species (from the specimens collected from Jinnah Garden, Lahore) suggesting improvement in the parameterization of the ibarcode reference library. This was especially true for the two most diverse orders (Hymenoptera: 8% and Diptera: 12%), where species assignments were less than 15%. Notably, similar studies conducted in other regions have achieved considerably higher assignment success rates; Canada (38%) and Germany (34%) (e.g., Geiger et al., 2016; deWaard et al., 2019a). Therefore, further optimization of the DNA barcode reference library (Lai, Li & Liu, 2023), updating DNA reference libraries with advanced software (Keck & Altermatt, 2023), and employing mini-barcodes for extensive species discovery and identification (Yeo, Srivathsan & Meier, 2020) may be recommended.

Although the reference database used in the current analysis was limited (Virgilio et al., 2010), the study still managed to identify representatives from a significant number of genera (324) and species (191). These findings highlight the advantages of using DNA barcoding over conventional morphological identification techniques since it is a more fast, reliable and accurate method (Marshall, Paiero & Buck, 2009; Antil et al., 2023). Furthermore, the analysis yielded 1,361 Barcode Index Numbers (BINs), indicating a high level of species richness in the fauna of Jinnah Garden in Lahore, Pakistan. However, it is crucial to emphasize that these estimates are derived from a restricted sample collection and limited geographic coverage, so more comprehensive efforts and broader sampling would likely result in even higher estimates of species richness.

Of the 17 arthropod orders identified in the study, five orders (Coleoptera, iDiptera, Hemiptera, Hymenoptera and iLepidoptera), from the class Insecta were the most abundant, collectively making up a substantial portion of 94% BINs. This finding corroborates earlier investigations employing both morphological i(Stork, 2018) and molecular techniques (Ashfaq et al., 2022; Pentinsaari et al., 2020). The dominance of Diptera and Hymenoptera can be attributed to the Malaise trap collection method, which preferentially captures low-flying insects, including these orders (Cooksey & Barton, 1981; deWaard et al., 2019b). Comparable patterns have been documented in other studies, such as those conducted in Canada where Diptera constituted approximately 57% of collections (deWaard et al., 2019b).

With 100 or more specimens, 16 of the 362 families dominated, and the BIN diversity mirrored this trend. The fact that 59 families were represented by a single BIN and 33 families by a single specimen suggests the survey’s inconsistent detection of families. It is interesting to note that of the 21 families with the highest BINs, eight were dipterans, and the greatest BINi. The specimen ratio was found in the family Cecidomyiidae (Fig. 4).

The analysis of species richness extrapolation, utilizing the iPreston log-normal species distribution model, suggest that a comprehensive sampling effort of the fauna at the Jinnah Garden has uncovered a significantly higher number of BINs than what has been observed to date. A similar results were documented by deWaard et al. (2019b). However, sampling method other than malaise traps is recommended to provide a more holistic understanding of the site’s biodiversity.

The BOLD houses an extensive database, and is a dependable platform for evaluating faunal overlap through BINs, with over nine million DNA barcode records for over 760,000 animal species. This study represents a significant advancement in establishing an inventory of the arthropod fauna in Lahore, Pakistan. However, the scope and quality of a relevant reference library is paramount in identification by means of DNA barcoding (Suriya et al., 2020; Huemer & Mutanen, 2022; Ramirez et al., 2023). Therefore, it is strongly recommended to develop the local biodiversity inventories and regional barcode libraries.

Supplemental Information

Supplemental Information 1 Count of specimens with DNA barcode records from Jinnah Garden in Lahore, Pakistan, that belong to 17 orders.

For every order, the number of families, genera, species, and BINs is given.

Supplemental Information 2 Bin Discordance Report.

Supplemental Information 3 Bin Discordance report of Singleton Bins.

Supplemental Information 4 CLUSTAL sequence analysis conducted by BOLD.

Supplemental Information 5 Taxon ID tree generated by BOLD.

Supplemental Information 6 Diversity measures analysis.

Additional Information and Declarations

Competing Interests

Author Contributions

Data Availability

The authors declare that they have no competing interests.

Khush Bakhat Samreen performed the experiments, analyzed the data, prepared figures and/or tables, authored or reviewed drafts of the article, and approved the final draft.

Farkhanda Manzoor conceived and designed the experiments, analyzed the data, authored or reviewed drafts of the article, and approved the final draft.

The following information was supplied regarding data availability:

The data and specimen details are available at BOLD: DS-GMPJA.

https://boldsystems.org/index.php/Public_SearchTerms?searchMenu=records&query=DS-GMPJA&taxon=.

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
