# Peer review of "Assessing arthropod biodiversity with DNA barcoding in Jinnah Garden, Lahore, Pakistan"

_PeerJ, doi:10.7717/peerj.17420_

## Round 0.1 · original submission · Major Revisions

Based on the the Reviewer's comments, your article needs Major revision. All three reviewers have also provided annotated manuscripts with extensive additional comments.

·

Basic reporting

The manuscript is well written and explained properly by the authors. English used in this paper is upto the mark. Literature references are sufficient to give background of the study. Article is professional with its structure. Results are relevant to the hypotheses.

Experimental design

The experimental design is well explained . Methodology is sufficient. Few suggestion are given in the text file,

Validity of the findings

The work is novel and rationale is clearly stated. The data have been statistically analyzed properly. Conclusions are well stated.

Additional comments

Format References according to format of Journal.

Reviewer 2 ·

Basic reporting

This article is a great contribution to the knowledge of Pakistan's diversity and in a biological group of great interest; it is generally well written, but it is recommended that the language be checked by an English speaker.
The abstract does not indicate all the relevant data nor all the analyses that were done (diversity estimation, number of Linnean taxa, etc.) so it is suggested that this could be improved.
In general, the figures and tables are sufficient, the resolution of the images is low in the PDF and they are well described, with minimal errors noted in the document. But it is suggested to increase a table with the comparison of the morphological and molecular determination at different scales of the taxonomic hierarchy for a better understanding. Several ideas throughout the manuscript in different sections lack citations to support such ideas. They are noted in the document.
It is necessary to describe the methods used, to make it replicable. I suggest increasing the literature that talks about both advantages and disadvantages of using this tool in understanding diversity. They are noted in the document.

Experimental design

The impact of the results will be a good contribution to the diversity of the country and the group. However, the work requires a greater description of the methods, metrics and criteria used for the generation and interpretation of the results. Specifically, the number of Linnaean names obtained on different scales is unclear, which is a crucial part with which the work compares its results.
In addition to this, several aspects of the methods do not allow replication, since they must be better described, such as the description of the sites, their location and selection criteria, the taxonomic determination of many arthropod orders, the values and criteria used in the different metrics. Other methods are not described although the results are mentioned, such as richness and diversity estimation, so this section needs to be improved substantially.

Validity of the findings

This article will provide a large amount of data that will be available in open access repositories, so it will be of great interest once the required information is completed.
Regarding the results found, although the authors offer a discussion about the limitations of the tool used, they do not really discuss the limitations of the work itself, since several aspects of the determination, of the biological group, of the collection methods used, as well as those of preservation, are limiting or have an effect on the results found. It is suggested that the authors offer a discussion section that focuses on the limitations of this specific research. It is also suggested to point out several of the limitations of the use of barcode, there is a lot of current literature that is not included (e.g. Suriya et al., 2020; Huemer and Mutanen, 2022; Ramírez et al., 2023).

Additional comments

The research is of great interest, as it focuses on a poorly studied group of animals; uses innovative tools and makes a great effort to compare, contrast and evaluate the results found with morphological knowledge; However, it requires improving all the methods and being more critical of the results achieved and the tool used.

Annotated reviews are not available for download in order to protect the identity of reviewers who chose to remain anonymous.

·

Basic reporting

Samreen and Manzoor present results of a very exhaustive work on barcodes in arthropods.

I consider it a valuable and significant contribution to understanding the diversity of the region and the role of molecular markers. However, the authors fail in the analysis of the results. Often overemphasizing the role of the barcode.

Experimental design

Finally, the authors should briefly present the morphology results, since they indicate that they used it to identify, but do not detail at what level.

Validity of the findings

The authors generally tend to simplify explanations, without taking into account factors inherent to the sequencing process (for example, that the primers were designed for certain groups).

They also do not consider natural wealth patterns and assume that differences in wealth and abundance are explained by collection methods.

There is confusion regarding the problems associated with successful sequencing and those that refer to problems in identification. The authors should address these issues separately.

Finally, the authors should briefly present the morphology results, since they indicate that they used it to identify, but do not detail at what level.

Additional comments

See file

---

## Round 0.2 · Minor Revisions

Dear Authors, Reviewer 3 has commented on the discussion; please address them as soon as possible.

·

Basic reporting

The manuscript is well written and the data provided is sufficient. Figures and tables are well presented.

Experimental design

Research methdology is properly explained

Validity of the findings

The data is valid

Additional comments

N/A

·

Basic reporting

No comment

Experimental design

no comment

Validity of the findings

no comment

Additional comments

The authors did a great job addressing the comments, so I have no problem with it being published.

I included a couple of comments in the discussion section.

---

## Round 0.3 · Minor Revisions

Reviewer 2. state that the manuscript does not have any incorrigible flaws, however, the errors that it still presents are relevant and should be corrected before it can be published. Language correction and justification of unsupported claims are fundamental. Please address Reviewer 2 comments.

·

Basic reporting

The manuscript is well written

Experimental design

The experimental design is very well explained

Validity of the findings

The data is valid

Reviewer 2 ·

Basic reporting

1. Basic reporting
1.1. The writing could be improved to clarify the meaning of certain sentences or paragraphs. Some information is also duplicated. Specific changes are suggested throughout the manuscript, but it is not an exhaustive review, so I recommend that it be reviewed by a fluent English speaker or a specialized language service. (Lines 62; 112; 127; 133; 135; 142-143; 179; 184; 191-193; 238; 244-245; 283; 311-312; 331; 335).
1.2. Some formatting errors were found in the manuscript. (Lines 87; 139; 147; 172; 215; 295; 316; 337)
1.3. The abstract is missing information, specifically the number of BINs recorded, which is a fundamental result of the paper.
1.4 The background, figures, raw data, and structure are adequate.

Experimental design

2. Experimental design
2.1. The question is clear and contributes to the knowledge of the diversity of the region.
2.2. The techniques used are adequate, both in the field and in the laboratory, as well as the data analysis.
2.3. The methods in general are adequate, however, there are some specific details missing that I comment on in the methods section of the manuscript.
2.4. If only malaise trap data were used, it is not necessary to mention the other collection methods (Lines 137-138; 154).
2.5. Be more precise about the duration of the sampling (Line 140).
2.6. Some of the numbers presented are confusing and it is not clear what they refer to when they are first mentioned. The meaning of some of them is clarified later, but not all of them. Please review the comments so that it is explicit in the manuscript what these numbers represent. (Lines: 32; 173; 224-225).
2.7. In the methods section, data are given that are part of the results, which is confusing because it is not explained what these numbers represent. I recommend limiting the methods section to describing the procedures. (Lines 173; 205; 216-217).

Validity of the findings

3. Validity of the findings
3.1. The results are valid, however, the manuscript could be improved by expanding the discussion on the limitations of the results. The authors focus on the virtues of this technique without mentioning or delving into its shortcomings (Lines 331; 336-337).
3.2. I suggest that practical recommendations based on the results be given for future studies, this would increase the value of the manuscript. The discussion reads like a description of the results when it could be more informative. I make specific suggestions on the manuscript. (Lines 306-308).
3.3. Some claims made throughout the manuscript are not supported by the results or the literature. In each of these claims, I comment in the manuscript why I consider them erroneous or need to be justified with theory or evidence.
3.3.1. On the greater reliability of determinations with molecular data compared to morphology and the high morphological variability compared to genetic variability (Lines 50-52).
3.3.2. On taxonomic determination and morphological convergence between closely related species (Lines 65-66).
3.3.3. On the impossibility of the COI gene to present homoplasy (Line 79).
3.3.4. On the orders with the highest diversity (Line 214).
3.3.5. On the interpretation of the proportions of identified species, it is mentioned that they are higher for insects compared to the total when they are lower (Lines 254-255).
3.3.6. On the fact that the diversity indices show a high equitability when the results of the abundance distribution suggest the opposite (Lines 270).
3.3.7. On the objective and scope of the study (Lines 277-278).
3.3.8. On the comparison of the % of DNA barcode recovery and the % of taxonomic determination with other studies (Lines 282-290).
3.3.9. On the fact that the % of taxa identified at the species level demonstrate that there is an improvement in the parameterization of the reference libraries (Lines 302-303).

Additional comments

4. General comments
The manuscript has value because of the question it answers, and the methods used are correct. These techniques for diversity quantification are relatively new, and any attempt at their implementation provides data to evaluate them.
In my opinion, the manuscript can still be improved and has some specific errors that need to be corrected before it can be published. Many of the doubts arise from the writing of the text, verb tenses, or word selection. The idea that the authors want to express is sometimes not clear. A language and format review would greatly improve the message and could clarify some of the comments I made.
Another important part of the corrections is to eliminate the results that are presented in the methods section, since this confuses the reader when trying to interpret data that have not yet been explained.
In addition, some assertions are unfounded, they are contrary to what the results show or do not have a theoretical basis. This does not significantly affect the conclusions, but it is necessary to justify or correct them.
Finally, the value of the manuscript could be increased by expanding the discussions with three aspects:
1) Analyze the shortcomings of this diversity quantification technique. For which groups is it useful? Are the results satisfactory? What are the weak points?
2) Identify the biases and limitations of this particular study.
3) After working in this field, the authors can make specific practical recommendations to improve the procedure from data collection to database use, or to improve the databases themselves.

Annotated reviews are not available for download in order to protect the identity of reviewers who chose to remain anonymous.

·

Basic reporting

I have reviewed the authors' response and the corrected version of the manuscript, I have no problem with it being published.

Experimental design

I have reviewed the authors' response and the corrected version of the manuscript, I have no problem with it being published.

Validity of the findings

I have reviewed the authors' response and the corrected version of the manuscript, I have no problem with it being published.

Additional comments

I have reviewed the authors' response and the corrected version of the manuscript, I have no problem with it being published.

---

## Round 0.4 · accepted · Accept

Authors have addressed all the reviewers comments. The article is ready to further process; publication etc